# Learning agents with prioritization and parameter noise in continuous state and action space

## Abstract

Reinforcement Learning (RL) problem can be solved in two different ways - the Value function-based approach and the policy optimization-based approach - to eventually arrive at an optimal policy for the given environment. One of the recent breakthroughs in reinforcement learning is the use of deep neural networks as function approximators to approximate the value function or q-function in a reinforcement learning scheme. This has led to results with agents automatically learning how to play games like alpha-go showing better-than-human performance. Deep Q-learning networks (DQN) and Deep Deterministic Policy Gradient (DDPG) are two such methods that have shown state-of-the-art results in recent times. Among the many variants of RL, an important class of problems is where the state and action spaces are continuous — autonomous robots, autonomous vehicles, optimal control are all examples of such problems that can lend themselves naturally to reinforcement based algorithms, and have continuous state and action spaces. In this paper, we adapt and combine approaches such as DQN and DDPG in novel ways to outperform the earlier results for continuous state and action space problems. We believe these results are a valuable addition to the fast-growing body of results on Reinforcement Learning, more so for continuous state and action space problems.

## 1 Introduction

Reinforcement learning (RL) is about an agent interacting with the environment, learning an optimal policy, by trail and error, for sequential decision making problems in a wide range of fields such that the agent learns to control a system and maximize a numerical performance measure that expresses a long-term objective (Li (2017)).

### 1.1 RL in Continuous State and Action Space

Many interesting real-world control tasks, such as driving a car or riding a snowboard, require smooth continuous actions taken in response to high-dimensional, real-valued sensory input. In applying RL to continuous problems, the most common approach for a long time has been first to discretize state and action spaces and then to apply an RL algorithm for a discrete stochastic system (Doya (2000)). However, this discretization approach has a number of drawbacks.

Hence, the formulation of the reinforcement learning problem with continuous state and action space holds great value in solving more real-world problems.

### 1.2 Deep Reinforcement Learning

The advent of deep learning has had a significant impact on many areas in machine learning, dramatically improving the state-of-the-art tasks such as object detection, speech recognition and language translation LeCun et al. (2015). The most important property of deep learning is that deep neural networks can automatically find low-dimensional representations of high-dimensional data.Therefore, deep learning enables RL to scale to problems which were previously intractable - setting with high dimensional/continuous state and action space Arulkumaran et al. (2017).

Few of the current state of the art methods in the area of Deep RL are:

- Deep Q-learning Networks (DQN) - introduced novel concepts which helped in using neural networks as function approximators for reinforcement learning algorithms (for continuous state space) Mnih et al. (2013).

- Prioritized Experience Replay (PER) - builds upon DQN with some newer approaches to outperform DQN (for continuous state space) Schaul et al. (2015).

- Deep Deterministic Policy Gradients (DDPG) - follows a different paradigm as compared to the above methods. It uses DQN as the function approximator while building on the seminal work of Silver et al. (2014) (for both continuous state and action space) on deterministic policy gradients .

### 1.3 PRIORITIZED EXPERIENCE REPLAY IN DDPG ( PDDPG)

We propose a new algorithm, Prioritized DDPG using the ideas proposed in DQN, prioritized experience replay and DDPG such that it outperforms the DDPG in the continuous state and action space. Prioritized DDPG uses the concept of prioritized sampling in the function approximator of DDPG. Our results show that prioritized DDPG outperforms DDPG in a majority of the continuous action space environments. We then use the concept of parameter space noise for exploration and show that this further improves the rewards achieved.

## 2 PREVIOUS WORK

### 2.1 CRITIC METHODS OF RL

Critic methods rely exclusively on function approximation and aim at learning a "good" approximation of the value functionKonda & Tsitsiklis (2000). We survey a few of the recent best known critic methods in RL.

### 2.1.1 DEEP Q-LEARNING NETWORKS

As in any value function-based approach, DQN method tries to find the value function for each state and then tries to find the optimal function. Min et al. Silver et al. (2014) also consider the approach of continuous state spaces. The novelty of their approach is that they use a non-linear function approximator efficiently. Until their work, non-linear function approximators were inefficient and also had convergence issues.

This was due to the fact that there was a lot of correlation between the data being fed to the neural networks which resulted in them diverging (neural networks assume that the data comes from independent and identically distributed source). To overcome this drawback, the novel ideas that were proposed which made it possible to use non-linear function approximator for reinforcement learning are the following:

- Experience Replay Buffer: In this technique, the input given to the neural network is selected at random from a large buffer of stored observations, which ensures that there are no correlations in the data, which is a requirement for neural networks.

- Periodic Target Network Updates: The authors propose that having two sets of parameters for the same neural networks can be beneficial. The two sets of parameters are used, one of the parameters is used to compute the target at any given iteration and the other, network parameters are used in the loss computation and are updated by the first network parameters periodically, this also ensures lesser co-relation.

### 2.1.2 PRIORITIZED EXPERIENCE REPLAY

The prioritized experience replay algorithm is a further improvement on the deep Q-learning methods and can be applied to both DQN and Double DQN. The idea proposed by the authors is as follows: instead of selecting the observations at random from the replay buffer, they can be chosen

based on some criteria which will help in making the learning faster. Intuitively, what they are trying here is to replace those observations which do not contribute to learning or learning with more useful observations. To select these more useful observations, the criteria used was the error of that particular observation.

This criterion helps in selecting those observations which help the agent most in learning and hence speeds up the learning process. The problem with this approach is that greedy prioritization focuses on a small subset of the experience and this lack of diversity might lead to over-fitting. Hence, the authors introduce a stochastic sampling method that interpolates between pure greedy and uniform random prioritization. Hence, the new probability if sampling a transition $i$ is

$$P(j) = \frac{p_j^\alpha}{\sum_k p_k^\alpha} \tag{1}$$

where $p_i$ is the priority of transition $i$ and $\alpha$ determines how much prioritization is used. This approach, while it improves the results has a problem of changing the distribution of the expectation. This is resolved by the authors by using Importance Sampling (IS) weights

$$w_i = \left( \frac{1}{N} \cdot \frac{1}{P(i)} \right)^\beta \tag{2}$$

where if $\beta = 1$, the non-uniform probabilities $P(i)$ are fully compensated (Schaul et al. (2015)).

## 2.2 ACTOR METHODS IN RL

Actor methods work with a parameterized family of policies. The gradient of the performance, with respect to the actor parameter is directly estimated by simulation, and the parameters are updated in the direction of improvement (Konda & Tsitsiklis (2000)).

### 2.2.1 DETERMINISTIC POLICY GRADIENTS (DPG)

The most popular policy gradient method is the deterministic policy gradient(DPG) method and in this approach, instead of having a stochastic policy which we have seen till now, the authors make the policy deterministic and then determine the policy gradient.

The deterministic policy gradient is the expected gradient of the action-value function, which integrates over the state space; whereas in the stochastic case, the policy gradient integrates over both state and action spaces. What this leads to is that the deterministic policy gradient can be estimated more efficiently than the stochastic policy gradient.

The DPG algorithm, presented by Silver et al. (2014) maintains a parameterized actor function $\mu(s|\theta^\mu)$ which is the current deterministic policy that maps a state to an action.They used the normal Bellman equation to update the critic $Q(s, a)$. They then went on to prove that the derivative expected return with respect to actor parameters is the gradient of the policy's performance.

## 3 ACTOR-CRITIC METHODS

Actor critic models (ACM) are a class of RL models that separate the policy from the value approximation process by parameterizing the policy separately. The parameterization of the value function is called the critic and the parameterization of the policy is called the actor. The actor is updated based on the critic which can be done in different ways, while the critic is update based on the current policy provided by the actor (Konda & Tsitsiklis (2000) Feinberg & Shwartz (2002)).

### 3.0.1 DEEP DETERMINISTIC POLICY GRADIENTS

The DDPG algorithm tries to solve the reinforcement learning problem in the continuous action and state space setting. The authors of this approach extend the idea of deterministic policy gradients. What they add to the DPG approach is the use of a non-linear function approximator (Lillicrap et al. (2015)).

While using a deterministic policy, the action value function reduces from

$$Q^\pi(s_t, a_t) = \mathbb{E}_{r_t, s_{t+1} \sim E}[r(s_t, a_t) + \gamma \mathbb{E}_{a_{t+1} \sim \pi}[Q^\pi(s_{t+1}, a_{t+1})]] \tag{3}$$

to

$$Q^\mu(s_t, a_t) = \mathbb{E}_{r_t, s_{t+1} \sim E}[r(s_t, a_t) + \gamma Q^\mu(s_{t+1}, \mu(s_{t+1}))] \tag{4}$$

as the inner expectation is no longer required. What this also tells us is that the expectation depends only on the environment and nothing else. Hence, we can learn off-policy, that is, we can train our reinforcement learning agent by using the observations made by some other agent. They then use the novel concepts used in DQN to construct their function approximator. These concepts could not be applied directly to continuous action space, as there is an optimization over the action space at every step which is in-feasible when there is a continuous action space (Lillicrap et al. (2015)).

Once we have both the actor and the critic networks with their respective gradients, we can then use the DQN concepts - replay buffer and target networks to train these two networks. They apply the replay buffer directly without any modifications but make small changes in the way target networks are used. Instead of directly copying the values from the temporary network to the target network, they use soft updates to the target networks.

## 4 Prioritized Deep Deterministic Policy Gradients

The proposed algorithm is primarily an adaptation of DQN and DDPG with ideas from the work of Schaul et al. (2015) on continuous control with deep reinforcement learning to design a RL scheme that improves on DDPG significantly. The intuition behind the idea is as follows: The DDPG algorithm uses the DQN method as a sub-algorithm and any improvement over the DQN algorithm should ideally result in the improvement of the DDPG algorithm. But from the above-described methods, not all algorithms which improve DQN can be used to improve DDPG. That is because some of them need the environment to have discrete action spaces. So, for our work, we will consider only prioritized experience replay method which does not have this constraint.

### 4.1 Prioritized DDPG algorithm

Now, the improvement to the DQN algorithm, the prioritized action replay method can be integrated into the DDPG algorithm in a very simple way. Instead of using just DQN as the function approximator, we can use DQN with prioritized action replay. That is, in the DDPG algorithm, instead of selecting observations randomly, we select the observations using the stochastic sampling method as defined in equation 1. The pseudo-code for the prioritized action replay is given in Algorithm 1.

This algorithm is quite similar to the original DDPG algorithm with the only changes being the way the observations are selected for training in line 11 and the transition probabilities are being updated in line 16. The first change ensures we are selecting the better set of observations which help in learning faster and the second change helps in avoiding over-fitting as it ensures all the observations have a non-zero probability of being selected to train the network and only a few high error observations are not used multiple times to train the network.

## 5 Results

The proposed, prioritized DDPG algorithm was tested on many of the standard RL simulation environments that have been used in the past for bench-marking the earlier algorithms.The environments are available as part of the Mujoco platform (Todorov et al. (2012)).

### 5.1 Mujoco Platform

Mujoco is a physics environment which was created to facilitate research and development in robotics and similar areas, where fast simulation is an important component.

---

**Algorithm 1** PDDPG algorithm

---

1: Randomly initialize critic network $Q(s, a|\theta^Q)$ and actor $\mu(s|\theta^\mu)$ with weights $\theta^Q$ and $\theta^\mu$.
2: Initialize target network $Q'$ and $\mu'$ with weights $\theta^{Q'} \leftarrow \theta^Q, \theta^{\mu'} \leftarrow \theta^\mu$
3: Initialize replay buffer $R$
4: **for** episode = 1, M **do**
5:      Initialize a random process $N$ for action exploration
6:      Receive initial observation state $s_1$
7:      **for** t = 1, T **do**
8:          Select action $a_t = \mu(s_t|\theta^\mu) + N_t$ according to the current policy
9:          Execute action $a_t$ and observe reward $r_t$ and observe new state $s_{t+1}$
10:          Store transition $(s_t, a_t, r_t, s_{t+1})$ in $R$               ▷ Storing to the replay buffer
11:          Sample a mini-batch of $N$ transitions $(s_i, a_i, r_i, s_{i+1})$ from $R$ from $R$ each such that - $iP(i) = p_i^\alpha/\Sigma_i p_i^\alpha$                      ▷ Stochastic sampling
12:          Set $y_i = r_i + \gamma Q'(s_{i+1}, \mu'(s_{i+1}|\theta^{\mu'})|\theta^{Q'})$
13:          Update critic by minimizing the loss: $L = \frac{1}{N}\sum_i(y_i - Q(s_i, a_i|\theta^Q))^2$
14:          Update the actor policy using the sampled policy gradient:

$$\nabla_{\theta^\mu} J \approx \frac{1}{N}\sum_i \nabla_a Q(s, a|\theta^Q)|_{s=s_i, a=\mu(s_i)} \nabla_{\theta^\mu} \mu(s|\theta^\mu)|_{s_i}$$

15:          Update the target networks:

$$\theta^{Q'} \leftarrow \tau\theta^Q + (1-\tau)\theta^{Q'}$$
$$\theta^{\mu'} \leftarrow \tau\theta^\mu + (1-\tau)\theta^{\mu'}$$

16:          Update the transition priorities for the entire batch based on the error
17:      **end for**
18: **end for**

---

This set of environments provide a varied set of challenges for the agent as environments have continuous action as well as state space. All the environments contain stick figures with some joints trying to perform some basic task by performing actions like moving a joint in a particular direction or applying some force using one of the joints.

### 5.2 EMPIRICAL EVALUATION

The implementation used for making the comparison was the implementation of DDPG in baselines (Dhariwal et al. (2017)). The prioritized DDPG algorithm was implemented by extending the existing code in baselines.

The following are the results of the prioritized DDPG agent as compared to the DDPG work (Lillicrap et al. (2015)). The overall reward - that is the average of the reward across all epochs until that point and reward history - average of the last 100 epochs on four environments are plotted. The y-axis represents the reward the agent has received from the environment and the x-axis is the number of epochs with each epoch corresponding to 2000 time steps.

As seen in Figure 1, the Prioritized DDPG algorithm reaches the reward of the DDPG algorithm in less than 300 epochs for the HalfCheetah environment. This shows that the prioritized DDPG algorithm is much faster in learning.

The same trend can be observed in Figure 1 for HumanoidStandup, Hopper and Ant environments. That is, prioritized DDPG agent learns and gets the same reward as DDPG much faster. This helps is in reducing overall training time. Prioritized DDPG algorithm can also help in achieving results which might not be achieved by DDPG even after large number of epochs. This can be seen in the case of the Ant environment. Figure 1 shows that DDPG rewards are actually declining with more training. On the other hand, Prioritized DDPG has already achieved a reward much higher and is more stable.

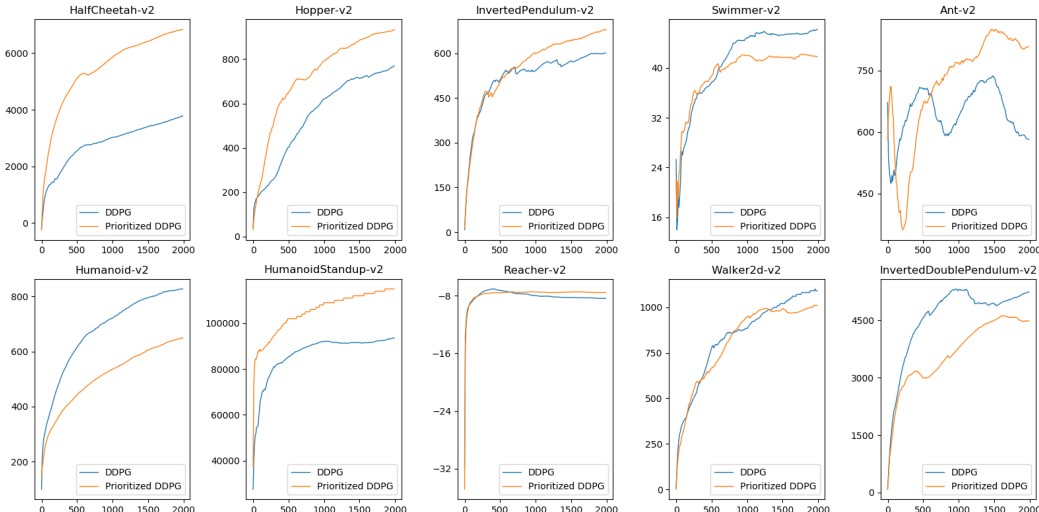

Figure 1: Prioritized DDPG vs DDPG

One a few environments such as the Reacher, InvertedDoublePendulum and Walker2d, it can be seen from Figure 1 the prioritized DDPG only ,marginally outperforms the DDPG algorithm.

# 6 PARAMETER SPACE NOISE FOR EXPLORATION

There is no best exploration strategy in RL. For some algorithms, random exploration works better and for some greedy exploration. But whichever strategy is used, the important requirement is that the agent has explored enough about the environment and learns the best policy. Plappert et al. (2017) in their paper explore the idea of adding noise to the agent's parameters instead of adding noise in the action space. In their paper Parameter Space Noise For Exploration, they explore and compare the effects of four different kinds of noises

- Uncorrelated additive action space noise
- Correlated additive Gaussian action space noise
- Adaptive-param space noise
- No noise at all

They show that adding parameter noise vastly outperforms existing algorithms or at least is just as good on majority of the environments for DDPG as well as other popular algorithms such as Trust Region Policy Optimization (Schulman et al. (2015)). Hence, we use the concept of parametric noise in PDDPG algorithm to improve the rewards achieved by our agent.

## 6.1 PDDPG WITH PARAMETER NOISE

The PDDPG algorithm with parameter noise was run on the same set of environments as the PDDPG algorithm - the Mujoco environments. The empirical results are as follows.

As we can see from figures 2,3 and 4 we see a great amount of variation on the reward achieved. We can infer that prioritized DDPG clearly works better for adaptive-param and corelated noise as compared to uncorrelated noise. This could be due to the fact that prioritized DDPG already explores faster as compared to DDPG and hence adding more randomness for exploration is not going to bear any fruit. Therefore, we can conclude that, PDDPG learns faster than DDPG and with the appropriate noise, it can be improved further. This can be seen in figure 5, where the overall best of both the algorithms have been plotted against each other. We see that PDDPG outperforms DDPG in majority of the environments and does reasonably well in the others.

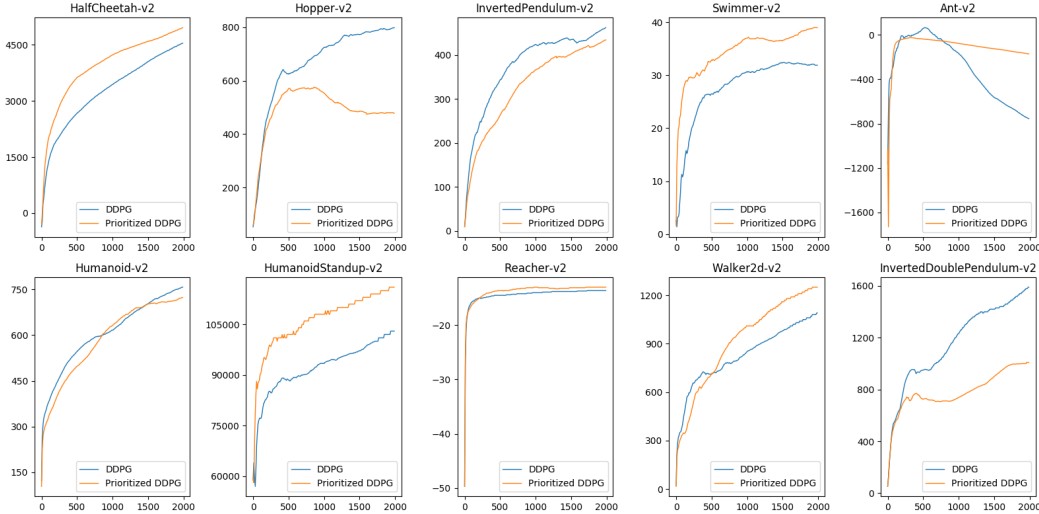

Figure 2: Prioritized DDPG vs DDPG with adaptive-param noise

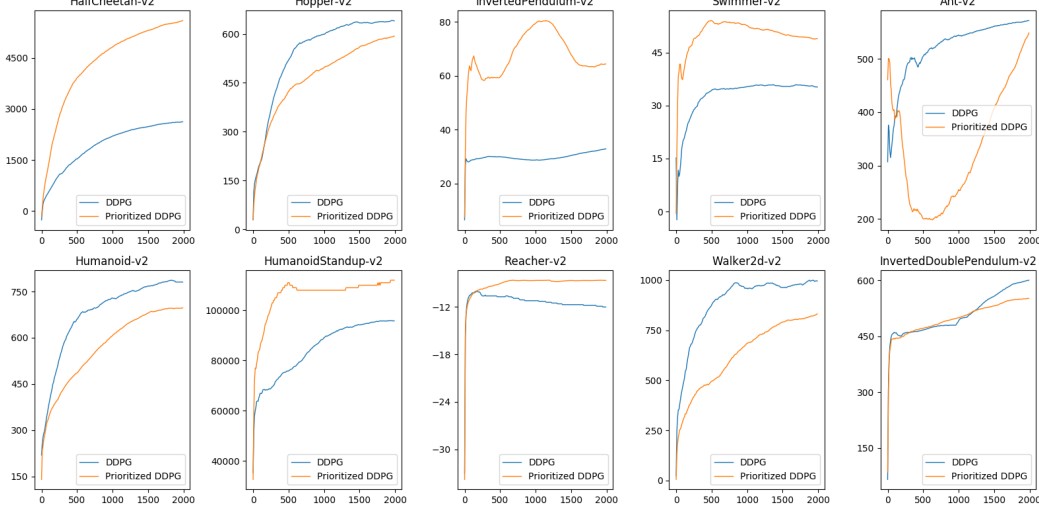

Figure 3: Prioritized DDPG vs DDPG with with correlated noise

# 7 CONCLUSIONS

To summarize, this paper discusses the state of the art methods in reinforcement learning with our improvements that have led to RL algorithms in continuous state and action spaces that outperform the existing ones.

The proposed algorithm combines the concept of prioritized action replay with deep deterministic policy gradients. As it has been shown, on a majority of the mujoco environments this algorithm vastly outperforms the DDPG algorithm both in terms of overall reward achieved and the average reward for any hundred epochs over the thousand epochs over which both were run.

Hence, it can be concluded that the proposed algorithm learns much faster than the DDPG algorithm. Secondly, the fact that current reward is higher coupled with the observation that rate of increase in reward also being higher for the proposed algorithm, shows that it is unlikely for DDPG algorithm to surpass the results of the proposed algorithm on that majority of environments. Also, certain kinds

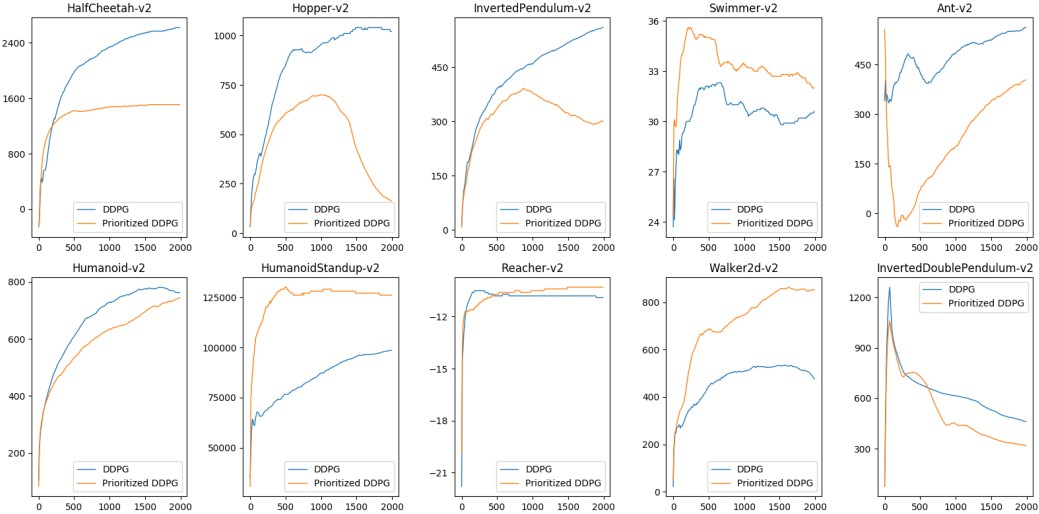

Figure 4: Prioritized DDPG vs DDPG with uncorrelated noise

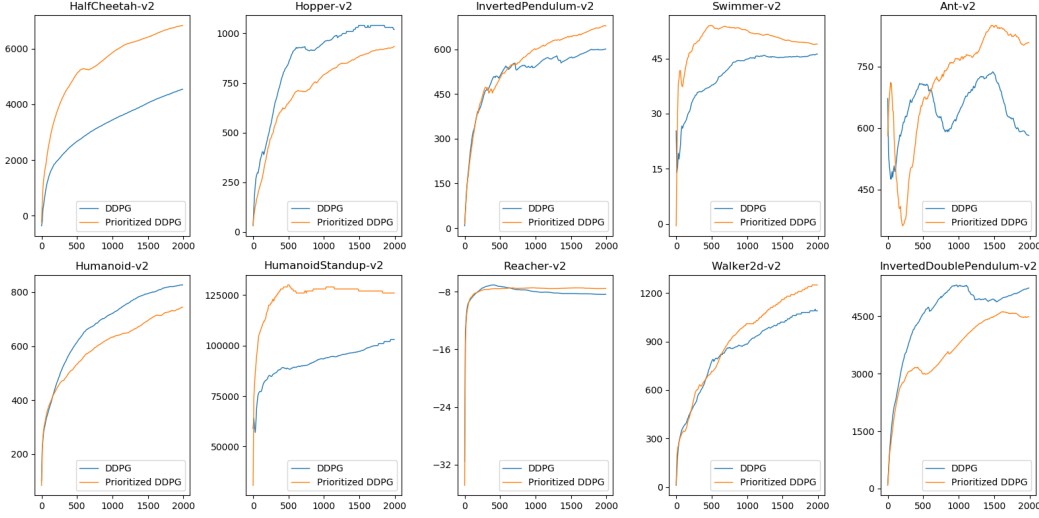

Figure 5: Prioritized DDPG vs DDPG with best results of both algorithms across all noises - adaptive-param, uncorrelated, co related and with no noise

of noises further improve PDDPG to help attain higher rewards. One other important conclusion is that different kinds of noises work better for different environments which was evident in how drastically the results changed based on the parameter noise.

The presented algorithm can also be extended and improved further by finding more concepts in value based methods, which can be used in policy based methods. The overall improvements in the area of continuous space and action state space can help in making reinforcement learning more applicable in real world scenarios, as the real world systems provide continuous inputs. These methods can potentially be extended to safety critical systems, by incorporating the notion of safety during the training of a RL algorithm. This is currently a big challenge because of the necessary unrestricted exploration process of a typical RL algorithm.

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
