# OpenReview forum: "Learning  agents with prioritization and parameter noise in continuous state and action space"
_ICLR.cc/2019/Conference_

### Official Review · AnonReviewer1 · 2018-10-28
**Limited novelty**

**Rating:** 4
**Confidence:** 4

**Review:**

The paper proposes PDDPG, a combination of prioritized experience replay, parameter noise exploration, and DDPG. Different combinations are then evaluated on MuJoCo domains, and the results are mixed.

The novelty of the work is limited, and the results are hard to interpret: sometimes PDDPG performs better, sometimes worse, and the training curves are only obtained with a single random seed. Also presented results are substantially worse than current state of the art (e.g., TD3, SAC).

---

### Official Review · AnonReviewer3 · 2018-11-02
**Interesting paper but limited novelty**

**Rating:** 4
**Confidence:** 3

**Review:**

This paper combines elements of two existing reinforcement learning approaches, namely, Deep Q-learning Networks (DQN) with Prioritised Experience Replay (PER) and Deep Deterministic Policy Gradient (DDPG) to propose the Prioritized Deep Deterministic Policy Gradient (PDDPG) algorithm. The problem is interesting and there is a nice review of relevant work. The algorithm has a limited novelty with a simple modification of the DDPG algorithm to add the PER component. Experiment results show improvements in certain simulation environments. However, the paper lacks insight on how and why results are improved on some settings while performing worse than the others. Detailed comments are as follows:

1. Algorithm 1 is not self-contained. Yes, I understand that it is a slight modification to DDPG with changes being Line 11 and 16. But p_i^alpha is not defined anywhere in Algorithm 1. How the transition probabilties are updated on Line 16 is also not clear to me.

2. It would be better if multiple simulation runs on the same experiment can be performed to have a more reliable display of performance.

3. Section 6 is on Parameter Space Noise for Exploration. This is not the authors' proposed work so it is strange to have a separate section here. In the end of Section 1, the authors wrote that "We then use the concept of parameter space noise for exploration and show that this further improves the rewards achieved." This seems to be a bold claim from the varying performance displayed in Figure 2-4. Similar to Comment 2, more simulation runs and statistical tests need to be conducted to support this claim.

---

### Official Review · AnonReviewer2 · 2018-11-05
**Barely any novelty**

**Rating:** 3
**Confidence:** 4

**Review:**

The paper proposes an augmentation of the DDPG algorithm with prioritized experience replay plus parameter noise. Empirical evaluations of the proposed algorithm are conducted on Mujoco benchmarks while the results are mixed.

As far as I can see, the paper contains almost no novelty as it crudely puts together three existing algorithms without presenting enough motivation. This can be clearly seen even from the structuring of the paper, since before the experimental section, only a short two-paragraph subsection (4.1) and an algorithm chart are devoted to the description of the main ideas. Furthermore, the algorithm itself is a just simple addition of well-known techniques (DDPG + prioritized experience replay + parameter noise) none of which is proposed in the current paper. Finally, as shown in the experimental sections, I don't see a evidence that the proposed algorithm consistently outperform the baseline.

To sum up, I believe the submission is below the novelty threshold for a publication at ICLR.

---

### Meta-Review · Area_Chair1 · 2018-12-12

**Confidence:** 5
**Recommendation:** Reject

**Metareview:**

The authors take two algorithmic components that were proposed in the context of discrete-action RL - priority replay and parameter noise - and evaluate them with DDPG on continuous control tasks. The different approaches are nicely summarized by the authors, however the contribution of the paper is extremely limited. There is no novelty in the proposed approaches, the empirical evaluation is inconclusive and limited, and there is no analysis or additional insights or results. The AC and the reviewers agree that this paper is not strong enough for ICLR.